# Research and Development of a COVID-19 Tracking System in Order to Implement Analytical Tools to Reduce the Infection Risk

**DOI:** 10.3390/s22020526

**Published:** 2022-01-11

**Authors:** Erik Vavrinsky, Tomas Zavodnik, Tomas Debnar, Lubos Cernaj, Jozef Kozarik, Michal Micjan, Juraj Nevrela, Martin Donoval, Martin Kopani, Helena Kosnacova

**Affiliations:** 1Institute of Electronics and Photonics, Faculty of Electrical Engineering and Information Technology, Slovak University of Technology, Ilkovicova 3, 81219 Bratislava, Slovakia; tomas.zavodnik@stuba.sk (T.Z.); tomas.debnar@stuba.sk (T.D.); lubos.cernaj@stuba.sk (L.C.); jozef.kozarik@stuba.sk (J.K.); michal.micjan@stuba.sk (M.M.); juraj.nevrela@stuba.sk (J.N.); martin.donoval@stuba.sk (M.D.); 2Institute of Medical Physics, Biophysics, Informatics and Telemedicine, Faculty of Medicine, Comenius University, Sasinkova 2, 81272 Bratislava, Slovakia; martin.kopani@fmed.uniba.sk; 3Department of Simulation and Virtual Medical Education, Faculty of Medicine, Comenius University, Sasinkova 4, 81272 Bratislava, Slovakia; 4Department of Genetics, Cancer Research Institute, Biomedical Research Center, Slovak Academy Sciences, Dubravska Cesta 9, 84505 Bratislava, Slovakia

**Keywords:** COVID-19, human interactions tracker, 2.4 GHz proprietary protocol, body temperature, wearable device, telemedicine

## Abstract

The whole world is currently focused on COVID-19, which causes considerable economic and social damage. The disease is spreading rapidly through the population, and the effort to stop the spread is entirely still failing. In our article, we want to contribute to the improvement of the situation. We propose a tracking system that would identify affected people with greater accuracy than medical staff can. The main goal was to design hardware and construct a device that would track anonymous risky contacts in areas with a highly concentrated population, such as schools, hospitals, large social events, and companies. We have chosen a 2.4 GHz proprietary protocol for contact monitoring and mutual communication of individual devices. The 2.4 GHz proprietary protocol has many advantages such as a low price and higher resistance to interference and thus offers benefits. We conducted a pilot experiment to catch bugs in the system. The device is in the form of a bracelet and captures signals from other bracelets worn at a particular location. In case of contact with an infected person, the alarm is activated. This article describes the concept of the tracking system, the design of the devices, initial tests, and plans for future use.

## 1. Introduction

Global interconnections bring today not only great benefits but also significant risks. One of them is the rapid spread of infectious diseases in the population, as we have seen recently. In December 2019, a novel coronavirus, now named severe acute respiratory syndrome coronavirus 2 (SARS-CoV-2), caused a series of acute atypical respiratory diseases. The disease was termed coronavirus disease 2019 (COVID-19) [1] and has become a pandemic. The number of deaths from COVID-19 is continuously rising. To date, 249,522,981 positive cases and 5,048,656 deaths have been confirmed worldwide. In Slovakia, which has 5,460,000 habitants, 506,795 positive cases were identified and 13,166 people died as a direct result of the disease [2].

The World Health Organization (WHO) has issued a series of recommendations to slow down the spread of COVID-19. In particular, they discussed the necessity to reduce social activity and maintain physical distance. Many countries began to implement social distancing and different forms of lockdowns [3]. However, in certain situations, it is impossible to maintain limited contact between people. The world continues to move, and everyone has to meet their living needs. The risk of infection is high in these interpersonal relationships. In particular, infected people who have not been positively identified and are asymptomatic carriers pose a risk [4]. In Slovakia and many other countries, the persons who have recently come into direct contact with a positively tested person are contacted by healthcare staff. If they are not vaccinated, they are subsequently sent to quarantine and undergo a polymerase chain reaction (PCR) test. The disadvantage of this early warning system is the incomplete capture of carriers. It is practically impossible to detect asymptomatic individuals who do not know they are transmitting the virus and warn anonymous foreign people who have met the infected person. Additionally, the current overloaded situation precludes this manual tracking and makes it almost impossible and prone to human error.

It is precisely due to these shortcomings that a suitable solution is sought with the help of appropriate telemedicine facilities. To support the fight against COVID-19, the Slovak Republic came up with a supporting specific research project. The Faculty of Electrical Engineering and Information Technology at the Slovak University of Technology in Bratislava and the Jessenius Faculty of Medicine in Martin, Comenius University, began to develop technical and methodological procedures to slow the spread of the disease and monitor positive patients or people exposed to the virus. Our aim was to develop an automated contact tracking system using portable devices, which allows more reliable tracking of anonymous social interactions and is attractive for people. People exposed to a positive person at a time when the person did not yet know about the virus infection will be additionally informed of the situation and level of threat and advised to take the necessary steps to prevent the transmission of COVID-19 and reduce the reproductive number.

The alarm (threat level) is displayed optically via RGB light emitting diodes (LEDs) in a bracelet. The user can also check their status using the web interface, and if the wearer pairs the bracelet with their phone, he or she may also receive an SMS notification. However, it must be clearly stated to the wearer that the bracelet does not check their current state of infection in any way. It only tracks contacts and helps to find at-risk contacts. It may be advantageous if these portable tracker devices also include sensors for monitoring human physiology. In particular, measuring body temperature can provide information about the disease and give an early indication of elevated body temperature. The person will be informed and can seek professional help for a test. In the initial stages, of course, we do not calculate the mass deployment of the tracking system. As in several other countries, the facility is planned to be deployed primarily in places with a high concentration of people where it is impossible to trace contacts, such as large factories and mass social events. Compared to other tracking devices, our design offers a few original solutions and benefits. The user can remain significantly more anonymous, and using the 2.4 GHz proprietary protocol is especially cheaper and more resistant to interference. The extended range of this signal can be used for contact tracing and mutual communication. This configuration required the development of its detection gateway, but the energy expended brought great benefits. The Nordic 2.4 GHz proprietary protocol was chosen because of its simplicity and low power consumption. It uses the same frequency and modulation as the Bluetooth Low Energy (BLE) protocol. The company Nordic has allowed direct access to the physical layer of the Bluetooth stack, which allows for simple, fast, and low-power communication protocols to be created. Through direct control of the packet size and transmission times of the radio transceiver hardware, we can achieve better control of packet handling and power management. The precise assessment of the timing of packets and their size by the microcontroller unit is used to determine the length and approximate distance of contact with another tracker. Firmware does not need to include the entire BLE software stack in order to use the proprietary protocol, which makes the firmware smaller and makes debugging and modification of the firmware easier. Restricting transmission times also allows for more devices to be transmitting simultaneously without interference occurring. Thanks to these systems, we can ideally protect the most vulnerable population groups, such as the elderly or chronically ill patients. Regarding personal data, we follow the basic principle of General Data Protection Regulation (GDPR).

## 2. Symptoms, Transmission, and Monitoring of COVID-19 Physiological Manifestations

COVID-19 is a respiratory disease affecting the human respiratory system directly. The SARS-CoV-2 virus produces clones in the human body, leading to the continuous transmission of the disease. Therefore, the isolation of infected individuals is essential for people’s lives in society [5,6]. The average time from exposure to onset of symptoms is 5 days; 97.5% of people develop the symptoms within 11.5 days [7]. Approximately 5% of COVID-19 patients and 20% of hospitalized patients experience severe symptoms requiring intensive care, including difficulty breathing, increase in heart and respiration rate, depletion of blood oxygenation (SpO_2_ level under 93%), severe cough, deviation in the electrocardiographic signal, and an increase in body temperature [8,9,10,11], as well as other symptoms. Overall, the symptoms of COVID-19 are extremely heterogeneous and dependent on the individual. Some patients have minimal symptoms while others develop worse symptoms, with some leading to acute respiratory distress syndrome (ARDS) with hypoxemia [1,12,13]. If presenting severe symptoms, the patient may become critical [14]. More than 75% of patients hospitalized with COVID-19 need oxygen supplementation. In Slovakia (updated data 19 October 2021), 23% of hospitalized patients have a difficult course, of which 52% are on oxygen supplementation [15]. The most common symptom of COVID-19 is increased temperature (70–90%), which we also focused on in the development of our device, followed by dry cough (60–86%), shortness of breath (53–80%), fatigue (38%), myalgia (15–44%), nausea/vomiting or diarrhea (15–39%), headache, weakness (25%), and rhinitis (7%) [16]. About 3% of patients experience loss of taste functions (ageusia) and loss of the ability to detect smells (anosmia) [7].

### 2.1. Ways of Spreading Diseases

The most beneficial protective step in COVID-19 disease is to not become infected. It is estimated that 48 to 62% of transmissions can occur via pre-symptomatic carriers. The virus spreads relatively quickly from an infectious person to another person. However, it is not clear how exactly the virus particles are transmitted and how infectious they are [17,18]. Epidemiological data suggest that the most common transmission is through the air during personal contact, especially small droplets of aerosols from talking, breathing, sneezing, coughing, or interacting with surfaces on which the drops have fallen [19]. The drops can be inhaled or spread through the mouth, eyes, or nose. Detection of viral nucleic acid in the air does not mean that the particles in the air are infectious [20]. Researchers have found viral ribonucleic acid (RNA) in air samples in hospital rooms used to treat people with both severe and mild COVID-19 disease [21]. However, none of the samples were infectious in cell culture experiments. When breathing without a cough, viral particles produced by patients may not always be viable. In another study, the researchers showed that the virus can stay in an aerosol for at least 3 h and still be infectious [22]. There are various controversies about the size of the droplets, the infectivity of the virus, and whether sick people produce enough infectious droplets or what the infectious dose is. The time and proximity of the infected person play an important role in the transmission of the virus. Brief contact (in 45 min) certainly produces enough particles to cause infection [18]. Long-term exposure (within 1.8 m for at least 15 min) and shorter exposure to individuals with symptoms are associated with a higher risk of transmission. Short exposures to an asymptomatic contact carry a lower risk [23,24]. Another study indicated a safe social distance of 1.6–3.0 m for large droplets from a discussion and up to 8.2 m considering all droplets. Some scientists also claim that room ventilation and various disinfection systems reduce the infection risk [25]. The second transmission route can be through touching surfaces where the virus droplets have fallen [14,21]. SARS-CoV-2 can remain on dry materials for several days under suitable conditions. Contact spreading through contaminated surfaces occurs to a lesser extent [19].

As the virus can spread rapidly, preventive instructions have been introduced worldwide, such as social distancing and the use of protective masks. However, their effectiveness is not 100%, and the virus is spreading despite epidemiological recommendations. The problem may be insufficient airway coverage with a material that provides an efficiency above 50% (e.g., cotton, natural silk, and chiffon) [26] or transmission through surfaces. There is an inverse relationship between humidity and ambient temperature and the length of virus survival on surfaces. It lasts well in an air-conditioned environment inside [27,28]. However, this may not be the case for all systems [29]. The virus is also stable on human skin, so hygiene and disinfection are crucial. SARS-CoV-2 can be quickly deactivated by UV radiation. The sun has a similar effect, and the pandemic is weaker in the summer as well [30]. These findings suggest that transmission occurs mainly indoors. It also happens because people inside are in closer contact, and therefore, the transmission of the disease is more prominent. Our device could, thus, find the best use inside.

### 2.2. Temperature as an Indicator of Human Health Condition

The most common symptom of COVID-19 is fever. Although it does not occur in all patients, it can help capture at least some patients. Elevated temperature is a non-specific marker of the whole spectrum of predominantly infectious diseases. From an evolutionary point of view, this is the most conserved response to an infectious stimulus. Interestingly, it is a very common symptom in COVID-19 (about 60% of patients in Slovakia and 70–90% worldwide [16]). The mean time for elevated body temperature is 10 days and correlates with clinical, laboratory, and radiological improvement.

To date, there are insufficient data to determine the type of temperature curve typical for COVID-19. Elevated body temperature at the beginning of disease during the virus replication phase is a physiological response to the onset of immunity. In cases of extreme inflammatory reaction, the fever reappears or continues, which is already counterproductive for the patient [31]. Overall, different authors have dissimilar attitudes to the temperature measures. The work of Schneider et al. [32] points out that fever is an insufficient indicator of the disease, mostly in younger age groups with mild to moderate disease (approximately 48% of cases, more often younger and women). Therefore, selecting individuals who develop fever may be important for faster diagnosis and subsequent isolation. Several papers [33,34,35,36,37] present the possibility of predicting mortality through temperature detection, so it is necessary to monitor and evaluate this parameter regularly, especially in intensive care units. However, it is not entirely clear whether achieving normothermia has a positive or negative effect in patients with a severe course of the disease. The onset of febrile illness caused by SARS-CoV-2 is unpredictable, and during the disease, the body temperature change develops individually for each patient.

Measuring the temperature of people outside the hospital in ordinary life is more difficult. Non-contact thermometers are used at the entrances to buildings where larger groups of people concentrate [38,39]. The study of Stave et al. [40] involved performing a number of screenings in the workplace and found that only one case out of 40 was detected. There may be variations due to different temperature screens when using instruments that measure the surface temperature of the skin and not the core temperature. Thermal camera measurement is challenging and inaccurate [41,42,43]. It has a larger measurement deviation than conventional thermometers and is sensitive to the environment. Moreover, activity of the human body can increase the temperature, but unlike the disease, it does not increase for a long time and can be easily recognized [42]. There are studies [44] that compare the accuracy of peripheral thermometers to central ones (catheters inside the body). As a result, peripheral thermometers do not have the required accuracy for the clinic but are commonly used in households and may find application in monitoring changes in a person’s temperature. The use of fever as a measured parameter is of little benefit [45], but it still identifies at least a certain number of carriers.

In general, temperature measurement and human monitoring could give better results than temperature measurement alone. Furthermore, the implementation of temperature measurement in the device is straightforward and low-cost.

## 3. State of the Art of Monitoring Devices

Several comprehensive solutions are available that differ in their approach to monitoring patients with COVID-19. Some of them are systems to collect data using control questions about the patient’s health and contacts from specialized applications sending data to the back-end database. From there, information is distributed to the healthcare provider [46]. The main benefit of this solution is eliminating the first doctor–patient contact, which directly prevents potential transmission to medical staff. The disadvantage is the patient’s potential misunderstanding or neglect of the control questions. This directly leads to misdiagnosis and prolongation of convalescence. It also does not allow real-time monitoring.

The next logical evolution is remote monitoring systems, where the vital signs are measured automatically and sent for analysis [47]. Systems on this basis consist of one-way communication, where measured data are sent to medical staff. Biometric devices worn on the body or implanted in a patient form a wireless network, as described in several scientific publications and works [48,49,50]. A more exciting project [51] shows a system that monitors the quarantined person’s body temperature, exercise activity, and indoor condition. However, despite the possibility of immediate detection of a change in the anamnesis, the declared system does not address the potential transmission of infectious disease. Furthermore, privacy and security issues related to COVID-19 prognosis and diagnosis are discussed through country case studies [52]. Yousaf et al. [53] explained the existing conflict between access to data and better services. They proposed permits for the collection and publication of COVID-19 patient data.

In addition, systems make it possible to identify an individual exposed to a disease transmitted by another person by safely monitoring interactions via smartphones between individuals that can be used during a public health emergency [54]. We also found two more interesting patents. The first [55] is a general model of various interactions adapted to the spread of diseases without a detailed description of the technical solution. The second one [56] is a proximity network map that defines who and what objects have come in contact with each other, including location and time. This map selects the list of people who have come into contact with known infected people based on contagious disease epidemiology criteria. Each person is carrying a proximity-sensing unit with a unique ID that records all other units encountered over time. Interactions are measured using a 900 MHz network, the exact location is recorded, and the movement of people is predicted. This system is considered mainly in terms of quarantine compliance. EIT Digital complements the smartphone-based approach with solutions based on physical tokens. Following a public call, EIT Digital received more than 60 expressions of interest. Thus far, they are active in four regions: Nordics, Benelux, Italy, and the UK. Token systems are easy to use, secure, and preserving and operate independently of mobile phones. All pilots have one thing in common: anonymity. Device wearers are anonymously notified if they have been in contact with a fellow wearer that has been infected with COVID-19. C-Detect is the name of the device in the United Kingdom [57]. It is attached to a bracelet and checks the respiratory rate, oxygen saturation, heart rate, and body temperature every 10 min. The first deployment is planned in hospitals. Crowdband has been developed in Benelux [58] and is also in the form of a wristwatch. The devices detect proximity and exchange anonymous IDs via broadcast radio. The bracelet system will be tested at football matches and major events such as concerts. In both cases, visitors will be offered to pick up and carry the equipment upon entry. Notification of other users in case of infection will remain voluntary. In Italy, the IprotectEU pilot project uses a bracelet-like token system. The initial phase will take place in a high school, followed by a factory, then an opera house, and finally a large concert venue. The Nordic pilot project is designed to operate in a hospital environment and on construction sites [51].

The alternative is using a decentralized network of devices (MESH network), working on the principle of proximity detection [59]. Interaction data are encrypted and can potentially be disseminated as a blockchain. The advantage is the security and anonymity of the system, but the potentially long time to spread data over this type of network is a major disadvantage. Moreover, due to the limited range of communication, such a system will become effective only with a higher percentage of devices in the population.

Many countries have also launched contactless smartphone solutions. For example, in German or French restaurants, it is possible to register at a table using the QR code for the place, and in the event of an infection in the relevant future, information on this risk will be provided. This ensures that the COVID-19 database for a particular restaurant contains only the actual guests who have arrived and the time stamp. Thus, all vulnerable persons will be informed without fail [60]. After scanning the QR code, the guest fills in all the necessary data in the form. The data are automatically transferred to the database and kept there for a maximum of 4 weeks [61].

Likewise, the Slovak Republic does not want to be left behind, and our project for an anonymous interaction tracking device (without a mobile phone) has started.

## 4. Design of the Tracking System

### 4.1. Principle

A tracking system has been developed to slow the spread of highly infectious diseases such as COVID-19 (Figure 1). For primary deployment, interior areas such as schools, hospitals, shops, post offices, large companies, and museums are considered. The main task of our facility is to monitor human interaction and, in the event of an infection, to track down the people who have been in direct contact with the infected person. In addition, measuring other physiological parameters will be a bonus and can make the device more attractive for future users.

The system is based on a wristband tracking device (Figure 2) that records contact duration and distance from other people wearing compatible trackers by measuring the received signal strength indicator (RSSI) and packet loss of a 2.4 GHz proprietary packet signal. The proprietary 2.4 GHz protocol was chosen for its low cost, lowest power consumption, ideal network topography, more effective data transfer, small memory usage, and long signal range [62,63]. However, in the 2.4 GHz frequency range, Wi-Fi networks are the primary source of interference. We chose to use the 2.4 GHz proprietary protocol and not Wi-Fi or Bluetooth because it uses different encoding and we can also change the wave modulation to minimize interference.

If one person becomes COVID-19-positive, the system warns people who have been exposed to increased contact with that person in the relevant past. The warning occurs either on the principle of light signalization through RGB LEDs directly on the tracker or via a secure web interface. The degrees of risk of infection are graded on the principle of traffic lights. All limit parameters such as contact duration and distance, length of tracking history, etc., are fully adjustable and can be configured according to the current instructions or hygiene standards. In the future, this will allow us to easily modify the system to consider an infectious situation, incoming mutations, and other infectious diseases. The limit parameters are set according to the recommendations of the Slovak public health authority for a contact distance of 2 m, a duration of 15 min, and a history of 14 days.

The tracking device also has a built-in thermal sensor that can continuously measure the temperature of the human body. After setting the body temperature limit and exceeding it, a notification is sent to the wearer. Body temperature, especially its changes, is a useful tool for monitoring the status of the infection. It is planned to estimate the correlation between changes in body temperature and the current state of COVID-19 infection.

The essence of the technical solution lies in the wireless proprietary 2.4 GHz protocol communication between individual tracking devices with a specific and unique identifier (ID). First, the personal tracking device scans all other system tracking devices it encounters and records contact distances, durations, and IDs to the internal memory nonstop. Subsequently, the scanned data are sent to the back-end database system and processed into the personal contact network of the individual system tracking devices. Finally, the data transmission between the tracking device and the back-end database system is realized anonymously using the same 2.4 GHz protocol at the sampling points in exposed places (gatehouses, entrances, elevators, etc.) via the detection gateway (Figure 2, middle). After mass expansion, it will be possible to use mobile phones as a collection gateway. Therefore, all hardware parts are ready for Bluetooth already. The tracking devices and the detection gateway are not equipped with a GPS module or any other form of position recorder. The tracking devices are also anonymized, and the data from the device are not distributed online.

The monitoring device is supported by the user interface and the database system (back-end), which provides a central computing point and uses extensive algorithms according to the data of experts such as epidemiologists, etc. Within the user interface, there is no link between the tracking device and the names of specific people. The database system (back-end) ensures the storage of personal data associated with the device using a unique identification code. An essential part of such a system is the security and encryption of personal data to prevent misuse by third parties and to ensure compliance with GDPR rules. Within this system, incoming data from tracking devices are also constantly collected and processed. This processing aims to create a network of personal contacts of individual tracking devices. The database system can use statistical tools to improve evaluation and reporting procedures. For better threat assessment, the contacts in this network can be divided according to the exposure level into any optional categories: moderate and high exposure categories. The first category includes indirect contacts with infected persons lasting less than *x* min and/or at a distance of more than *n* meters. The second category includes contacts with an infected person lasting more than *x* min and/or at a distance of less than *n* meters. The parameters *x* and *n* are adjustable values. These outputs, in the form of a network of contacts, can be used by the early warning method. In the event of a change in the status of a specific identifier, the system automatically evaluates the output in the form of a database of tracking device carriers exposed to close contact with the disease. To these people’s tracking device, a notification and distance data will be sent, and the contact time will be anonymously accessible from the user environment.

The network of contacts is also accessible in its anonymous form (Figure 3), after entering the unique hardware ID, within a web application that serves as an interface for the user and can fully provide extensive data, including identifying infection risks. This part is very important, especially for people who wear trackers, such as workers in a large company where the tracking device cannot be directly connected to a specific person. Pairing a mobile phone with a specific tracking device can be performed using a QR code or near field communication (NFC).

### 4.2. Hardware Design and Technical Specification

The heart of the tracking device (Figure 4, Table 1) as well as the detection gateway (Figure 5, Table 2) is the nRF52840 microcontroller unit (Nordic, Oslo, Norway). The nRF52840 unit is built around a 32-bit ARM^®^ Cortex™-M4 processor with a floating point unit running at 64 MHz. It has 1 MB Flash, 256 kB RAM, and protocol support for Bluetooth LE, Bluetooth mesh, Thread, ZigBee, 802.15.4, ANT, and 2.4 GHz proprietary stacks. In our application, we use the latter 2.4 GHz proprietary protocol for mutual communication and tracking. For the tracking device, this is performed specifically with the BL654 (Laird Connectivity, Akron, OH, USA) chip, and for the detection gateway, the BL654-PA chip, where the communication range is extended via a low-noise amplifier (LNA) and dipole RF antennas. The maximum communication distance in an ideal open space is 1.6 km, but from a real test inside a building, we can guarantee a fully sufficient 50 m. The stored data are immediately transferred if the tracker approaches the detection gateway zone.

The Laird BL654 tracking device in an actual configuration is capable of detecting other trackers in a mutual distance of 2.5 ± 1.5 m. In our case, the distance measurement is specifically based on a combination of the received signal strength indicator (RSSI) and a number of packet losses [64,65]. Packet loss describes packets of data not reaching their destination after being transmitted across a network. In our system, we know the number of packets sent per second. Packet loss in our case mostly depends on the distance and location of the people. By combining two parameters, we increase the reliability of the system. If one of the given values is out of range, the persons are at a safe distance, the packets are discarded, and the contact is not recorded. These properties can be changed during debugging. Contact time is an adjustable parameter at compile time. Thus far, this is conducted so that the tracker looks around every minute and saves the IDs of all trackers whose RSSI and packet loss meet the conditions. Once every 4.5 min, it then filters out those it has seen at least twice and marks them as a contact.

The built-in thermometer is SI7051-A20-IM (Silicon Labs, Austin, TX, USA). The SI7051 thermometer is based on a band-gap temperature sensor element and offers an accurate, low-power, factory-calibrated digital solution ideal for measuring temperature in telemedical applications. In the range of human body temperatures (35.8–41 °C), it offers an excellent accuracy of ±0.1 °C, or 0.13 °C at a range from 20 to 70 °C. The sensor is connected to the human body via a plate of surgical steel placed on the bottom part of the tracker.

During the initial phases, we found insufficient accuracy of the internal real-time clock (RTC) and therefore expanded the tracker with an external RV-8263-C7 RTC (Micro Crystal, Grenchen, Switzerland). The selected RTC has a drift of 10 s per month on average and its own backup battery in case the main battery has run out so the device can keep track of the records. The tracker can store acquired data in internal storage. Specifically, the 16 Mb model MX25R1635FZUILO (Macronix, Hsin-chu, Taiwan) works at 33 MHz clock speed. Note that about 200,000 meeting records can be stored on a 16-megabyte memory. The tracker’s functionality is complemented by RGB LED signalization. The power for the device is provided by a 300 mAh Li-Pol battery, which gives us an estimated endurance of 80 days. Switching on/off is provided by the device unplugging/plugging into the micro-USB charger. Recharging to full condition takes 60 min. There is also a service button on the device. It is currently used to verify the tracker’s status (via LED signalization) and force the sending of the recorded data. The tracker has an ergonomic design with small dimensions. Without a strap, the tracker body has a diameter of 46 mm and a height of 13 mm. The total weight (strap included) is 30 g.

In addition to the mentioned data collection from the tracker entering its signal zone using the proprietary 2.4 GHz communication protocol of the BL654-PA sensor, the detection gateway must, of course, ensure the transfer of these data to the back-end and database system. The data collected to the back-end can be sent via Wi-Fi using an ESP32 WROOM chip (Espressif Systems, Shanghai, China) or, in the absence of a Wi-Fi signal, via an LTE network thanks to an EG912Y chip (Quectel, Shanghai, China). Espressif’s ESP32 WROOM is a powerful low-energy Wi-Fi + Bluetooth/Bluetooth LE module targeted for a wide variety of IoT applications. The Quectel EG912Y module is optimized especially for machine-to-machine (M2M) and IoT applications with LTE, GSM/GPRS/EDGE data transmission.

The role of the detection gateway is not to store the collected data but to send them instantly via an Internet connection to the server for further processing. However, in case of signal failure, we installed 64 Mb MX25R6435FZNIL0 internal memory (Macronix, Hsin-chu, Taiwan), which is flash memory running at 80 MHz. The device dimensions are 105 × 105 × 27 mm (without antenna). The total weight without antennas is 100 g. The gateway power supply is 5 V/4 A via a power jack. The device can serve about 10 to 100 trackers at one time, depending on the type of operation. We achieved a total transmission of approximately 100 meeting records per second. In detection gateway overload, the customer should understand that he or she must equip the premises with an additional number of gateways.

### 4.3. Back-End and Software

A key element in establishing a reliable telemedicine system to support the tracking of the possible spread of infections is to build a strong background for the transmission, storage, and evaluation of large amounts of data regarding their sensitivity. The basic back-end configuration was prepared and tested, and the software protocols were set (Figure 6). The future implementation is beyond the scope of this paper.

The back-end manages information about the tracking devices belonging to it and receives, records, and evaluates interactions between the users of the trackers while continuously creating a network/graph of interactions. For example, suppose a user reports a relevant infection. In that case, it searches for the contacts in the interaction network and creates an alert for the affected users via LED signalization on specific trackers, SMS, or e-mail.

In detail, the back-end server using the application programming interface REST API (RedHat, Raleigh, NC, USA) collects individual messages from all detection gateways and replicates the entire contact report from tracker devices. Communication runs via https using a self-signed certificate. Individual recordings from trackers are base64-encoded. This report is transmitted in binary form, then decrypted and inserted into the PostgreSQL primary database. Authorization to the back-end for the detection gateway is realized by the OAuth2 bearer token. There are four columns of information in the database for further processing: the ID of the device that saw the contact, the ID of the device that was seen, the body temperature, and the time it was seen in UNIX timestamp format. The main part of the server is programmed in Python, and the computational part in C++. The open-source Apache Kafka platform is used as the streaming platform and for data integration of the current tracking data. Prometheus with the graphic superstructure Grafana is used as the monitoring solution. The website is served via an Nginx web server. Client access to the back-end and the creation of monitoring reports are provided by the API/Web Socket protocol.

An example of web access to the back-end can be seen in Figure 7. The preliminary graphic user interface (GUI), called IOT Health, provides the user a direct and efficient interpretation of data. It is integrated into the individual modules and ensures the normal operation of the system. IOT Health is our own product used to manage telemedicine equipment. It allows access to our electrocardiogram (ECG) holters, thermal “coins”, and just presented tracking device. Figure 7a shows active devices, and Figure 7b shows a 24-h record of a person’s peripheral temperature as measured by the tracker.

### 4.4. Functional Testing

To date, the first 18 prototypes of a tracking device and four detection gateways have been produced and tested. The first practical testing took place in the building of our school for 3 weeks. The users of the 15 trackers were our colleagues. The participants also kept paper records of the meetings (their duration and approximate distance) and their overall activities. These were compared with the hardware records. The relevant nine-floor building area was about 7000 m^2^, and the detection gateway was located at the gatehouse (floor 1). The back-end server was placed in the laboratory on the sixth floor. Written informed consent was obtained from each person in the primary testing.

## 5. Results of Functional Testing

We managed to build a functional tracking system. The temporary user interface from the real operation can be seen in Figure 8. In individual lines, the respective encounters (at a distance of less than 2 m) were recorded. At this stage, the trackers were still assigned to specific people. For example, in Figure 8, the tracker assigned to “Michal Micjan” can be seen. The specific record is from 18 March 2021, and there is a mark showing at what time he was in close interaction with which person. After clicking on the corresponding interaction, its details will appear. For example, the interaction with “Miro Novota” is highlighted here (green color). It starts at 11:36 and lasts 110 min. Whether the interactions were recorded correctly could be verified by checking the “Miro Novota” tracker. Logically, the interaction should be recorded equally on both trackers. From the overall results, we can see 97% agreement. Minor inconsistencies in the introduction were caused by a malfunction of the internal RTC as time shifts of interactions arose.

Figure 9 shows an example of the overall interaction analysis of 10 people over 5 days. As can be seen, there were two working groups. The first group, consisting of Miro, Vrato, Miso, Juro, and Adam, was on the second floor. The second group, consisting of Tomas, Krisztian, Jozo, Lubos, and Ricsi, was located on the fourth floor of our university. The graph shows that “penetration” contact between these groups was minimal. Miso had the most “penetration” contact between the groups because he works as a project manager and has to meet them all. Significant contacts can also be seen between Miso and Miro, Adam/Vrato, and Jozo/Lubos/Tomas/Ricsi because they share a common office and between Krisztian/Ricsi, who are brothers and ride to work together. In reality, neural networks will be deployed on similar models, and the main output is still interactions in terms of COVID-19 propagation. Contacts were recorded realistically, and they were not disturbed by Wi-Fi interferences.

Figure 10 shows an example of a 24-h temperature record obtained by one tracker device on 30 November 2021. The graph shows the basic events of the person with the temperatures of specific environments. These data were obtained from paper notes and serve to justify the temperature changes of the examined person. As we expected, the graph shows that the measurement of the wrist’s temperature depends on the surrounding conditions and specific physical activities. Therefore, it is planned to add another thermometer to the device to record the outdoor temperature, allowing us to evaluate the thermal gradient of the person better and obtain a more realistic value of the core body temperature. This will be evaluated internally in the device itself. Long-term temperature change trends will be significant. In the final analysis, neural networks, which are the main task in the project’s next phases, will again play an important role.

During the test, various issues were identified, and intensive improvements were made to the device. We found that the practical range of the detection gateway towards the trackers is about 70 m, so we can guarantee a 50-m detection gateway zone. In the early stages, we also found problems with the reliability of the internal RTC, so the tracker hardware was expanded with an RTC extruded circuit RV-8263-C7 with a backup battery, as mentioned in Section 4.2. Communication from the detection gateway to the back-end took place alternately via Wi-Fi and 4G LTE. The tracking of contacts and people’s temperature was completely reliable.

## 6. Discussion

The hardware design and primary testing were successful. The tracking devices, detection gateways, and the basic structure of the back-end worked reliably and met all requirements. The actual device can detect other trackers at a mutual distance of 2.5 ± 1.5 m. Although this specification may seem inaccurate, it is based on long-term testing in real-life situations and gives an average distance limit of 2 m. It is necessary to realize that in real life, people do not stand side-by-side passively, and their positions also change. They can stand upright, and the trackers will be in direct open contact with the strongest signal, but if they have their hands behind their backs, the signal will pass through two bodies, making it significantly weaker. They can sit in a restaurant, and the signal will pass through the tabletop. They may also have their arms outstretched, in which case the real distance between the bodies will be greater than it appears from the tracker signal. Thus, real calibration is not possible; it may thus be useful to define calibration protocols in the future. Our calibration protocol was performed by empirical measurements. We deployed the equipment to 15 people who were gradually placed at a distance of 2 m, where they rotated 360°, and packet loss and RSSI were evaluated and averaged. Unlike most of the devices presented in Section 3, our tracking device does not use common networks such as Bluetooth Low Energy (BLE) and near field communication (NFC) in smartphones to track contacts [54]; instead, we have chosen the 2.4 GHz proprietary protocol. This allows us to achieve lower costs, increase communication reliability in a highly disruptive environment, and stay anonymous. The interference problem did not occur during the entire testing period in an environment with strong 2.4 GHz RF network coverage. The range of our network is theoretically up to 1.6 km, so we can use the same network to monitor the distance of contacts and communicate out to the collection gates. In most of the presented devices, specific technical data were lacking, and the vast majority completely neglected collection points, which in our opinion are an integral part of such a system. Many of the described systems are only in the form of patents [54,55,56] without any hardware implementation. The systems closest to our system with a more detailed description of the technical implementation are in the European region associated with EIT Digital [51,57,58].

Some systems also try to integrate biosignal sensors into tracking devices, which we consider to be a great idea. Our current device incorporates a temperature sensor. During the test, we achieved credible results, especially in relatively constant conditions. Our system relieves staff and reduces the patient’s burden by measuring the temperature without the need for another person and awakening the patient. A significant advantage is the possibility of obtaining a statistically larger amount of data and monitoring the development of temperature trends. It is estimated that strong fluctuations in higher temperatures are associated with impaired regulation of the body during COVID-19 disease. In addition to the thermometer, we would like to mention the built-in DPS310 barometer (Infineon Technologies, Neubiberg, Germany) in the diagram. At present, we mention it only briefly because it is still in test mode, and the deployment is being considered. In the near future, the tracker will be further upgraded by a photoplethysmography (PPG) sensor to detect heart rate and blood oxygenation, which provide interesting data with respect to COVID-19. In this paper, we have not dealt in detail with wearable monitors of human physiology. There are already thousands of given products and a nice breakdown can be found, for example, in our publications [66,67]. However, our device has a few original solutions and benefits. The user can remain significantly more anonymous than if using a mobile phone paired with a specific person. We also do not determine the user’s location. We primarily look for a bracelet that shows the degree of threat. Compared to mobile phones, our device also has a significantly longer battery life. Many people, especially the elderly, do not know how to use or do not have mobile phones. Employers also cannot force people to install the software on their private phones. Furthermore, in many operations, the use of mobile phones or cameras is prohibited.

Many of the imaginary competing facilities lack a sufficient description of the technical solution. Interestingly, we did not find any single device using the 2.4 GHz proprietary protocol. By using it, our device can be cheaper and more resistant to interference and we have a greater signal range, so we can solve the monitoring of mutual distance and communication with only one signal. We made a detection gateway—something similar to a Wi-Fi router but just communicating on our set proprietary protocol. In parallel with the production of the tracker, we are also developing wearable health electronics, which has huge benefits at this time of the COVID pandemic. We plan to combine these solutions into one and thus gain further attractiveness for future users.

The second phase, which shall come soon, is the complete development of the back-end and software applications and additional hardware optimization. These activities will result in neural networks and machine learning focused on automated evaluation, screening, filtering, and prediction of behavior for the collected data. We will create predictive models of disease spread based on input data from actually tested samples. An interesting result may be markers of changes in body temperature for specific groups of the population in relation to the development of the disease, proving the dependence of the transmission of the disease on the degree of social interaction. The tracking system is not necessarily linked only to the current COVID-19 disease; it can be used in the future for a wide range of infectious diseases such as Crimean–Congo hemorrhagic fever, Lassa hemorrhagic fever, Rift Valley fever, Ebola virus, Marburg virus, Nipah virus, Middle East respiratory syndrome, severe acute respiratory syndrome, etc.

## 7. Conclusions

Here, we introduced the primary development phase of a telemedicine system to support the anonymous tracking of possible COVID-19 spread. We showed the hardware design of the tracking devices and collection points (detection gateways) and introduced the basic concept of the back-end system. After the implementation of appropriate evaluation algorithms and neural networks, the system will be a powerful tool to suppress the spread of infectious diseases. The methodology was granted a utility model. In the future, we plan to expand the system with other sensors of human physiology such as heart rate and blood oxygenation. After their implementation, a certain version of the holter will be actually created. An added measurement device of human physiology can make the device even more attractive for individuals. As a result, people will become interested in the device, and the need to convince people of the benefits of the product will be reduced.

## 8. Patents

The tracking system was granted a utility model by the Industry Property Office of the Slovak Republic under application number 50130-2020; Utility Model Number: 9322.

## Figures and Tables

**Figure 1 sensors-22-00526-f001:**
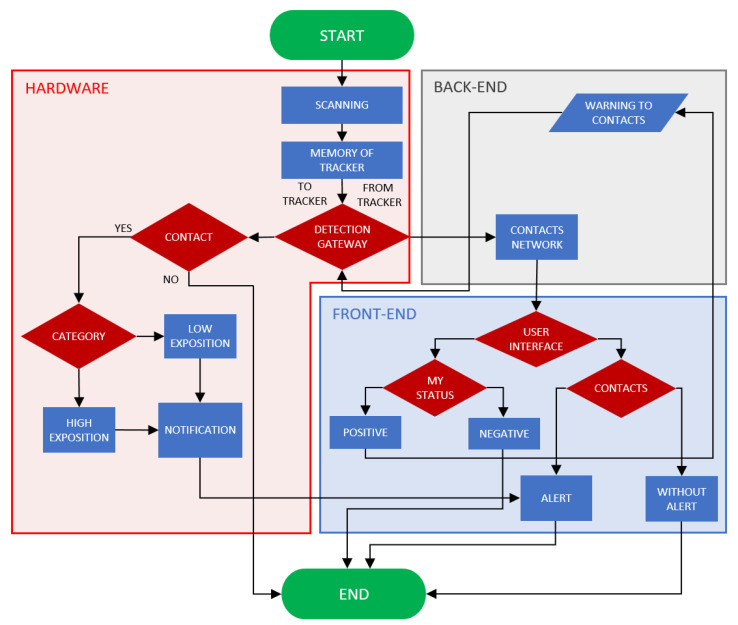
Functionality of proposed tracking system: Hardware—logical block, which declares the elemental principle of detection, communications with the server, and users’ notifications; Back-End—software part which consists of server-hosted network of contacts and server processing of warnings; Front-End—software logical block of the functions of user interface (UI).

**Figure 2 sensors-22-00526-f002:**
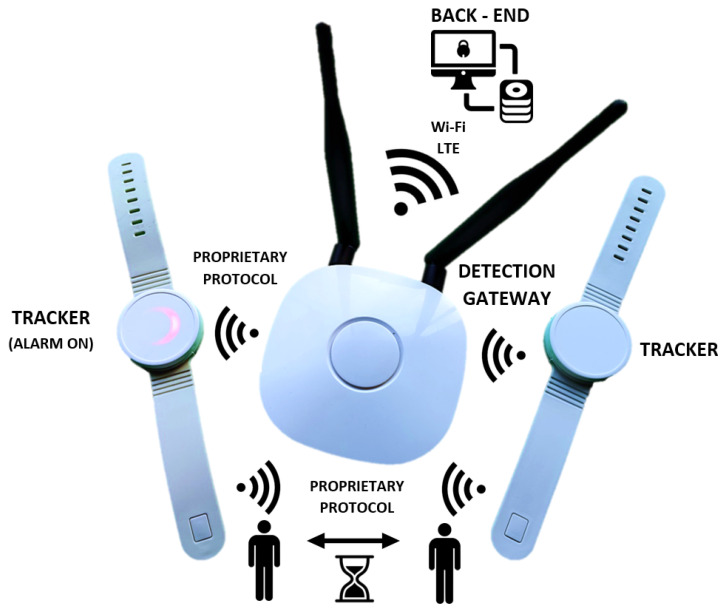
Designed hardware in the tracking system: (**left**/**right**) tracking device with alarm on/off tracking the length and duration of the mutual contact; (**middle**) detection gateway collecting data from trackers and communicating with the back-end.

**Figure 3 sensors-22-00526-f003:**
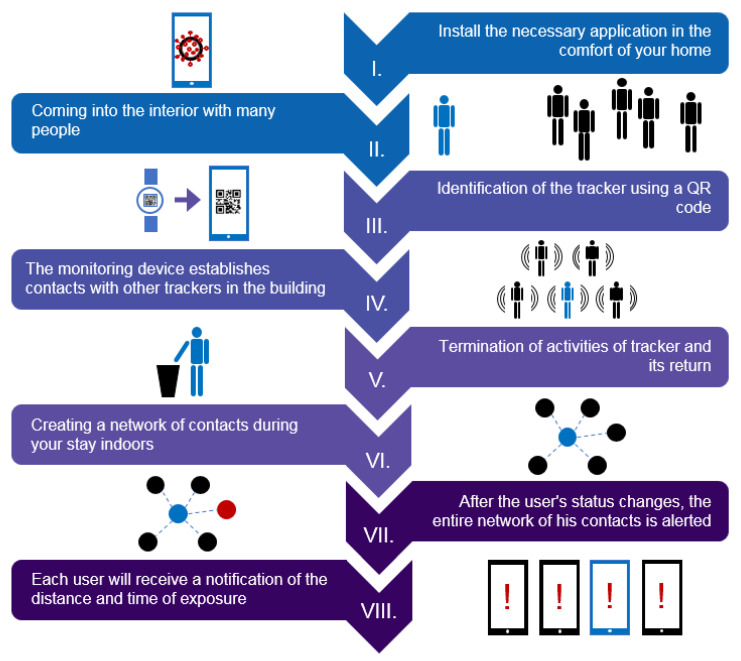
User interaction: linking a mobile phone to a tracking device. Graphical interpretation of the use case of the tracking device in a specific area, such as automotive facilities, shopping centers, etc. The use case includes fundamental features of the proposed system. All processes can be described as follows: (I) log in to the web application; (II) reach the monitored space; (III) link the web application to the tracking device; (IV) the tracking device establishes contact with other trackers in the area; (V) termination of activities and return of the tracking device and simultaneous (VI) creation of a network of contacts, which can be used if (VII) users change own status; and finally, (VIII) a notification is sent.

**Figure 4 sensors-22-00526-f004:**
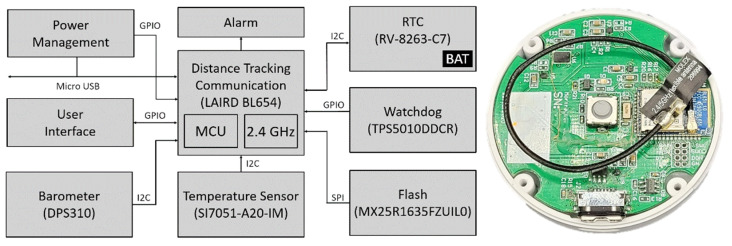
Tracking device block diagram and realized printed circuit board (PCB): Power management—integrated circuit (IC) responsible for charging the battery and maintaining a stable supply of voltage for the microcontroller unit (MCU) and other circuit logic; Alarm—RGB LED signalization; RTC—time recorder; Watchdog—component for resetting the system in case of MCU error; Barometer—atmospheric pressure monitoring; Temperature sensor—wrist temperature monitoring; Flash—long-term non-volatile storage for the MCU; User Interface—buzzer and button to communicate status, battery level, and syncing operation to the user; LAIRD BL654—main MCU used to control the connected components and direct operations of the device.

**Figure 5 sensors-22-00526-f005:**
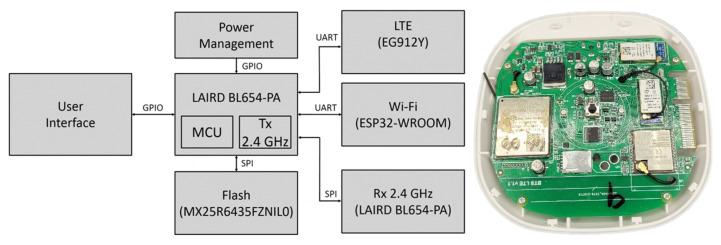
Detection gateway—block diagram and realized printed circuit board (PCB): Power management—integrated circuit (IC) responsible for charging the battery and maintaining a stable supply of voltage for the MCU and other circuit logic; LTE/Wi-Fi—back-end communication; Rx 2.4 GHz—communication with tracking devices; Flash—long-term non-volatile storage for the MCU; User Interface—communication of status and syncing operation to the user; LAIRD BL654-PA—main MCU used to control the connected components and direct operations of the device.

**Figure 6 sensors-22-00526-f006:**
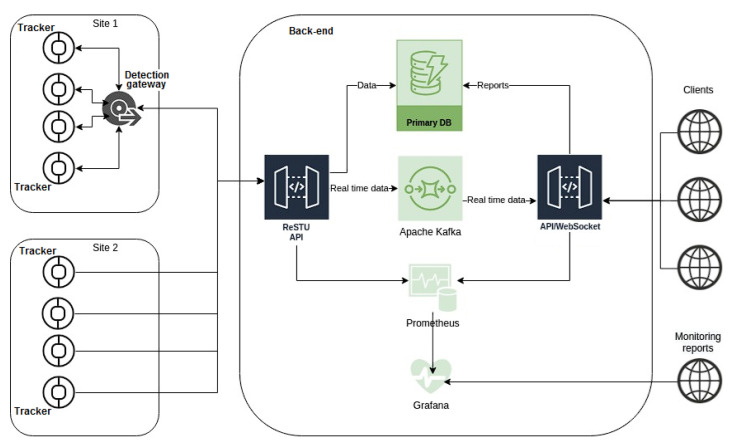
Back-end system configuration: Devices and users with communication lines and back-end system/applications organization.

**Figure 7 sensors-22-00526-f007:**
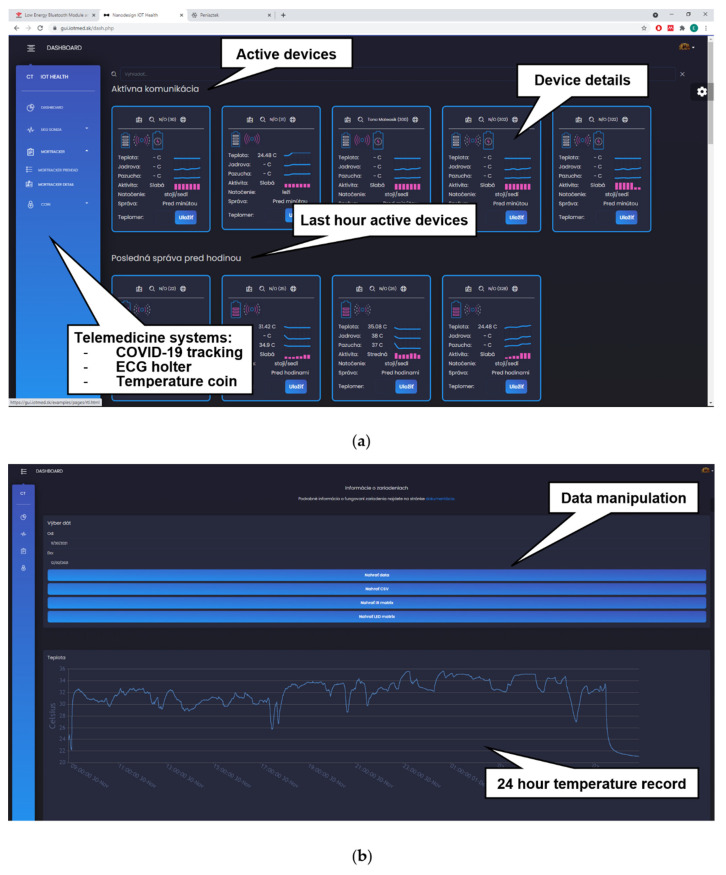
Web access to the back-end: (**a**) IOT Health system with displayed active devices; (**b**) daily graph of the measured peripheral temperature.

**Figure 8 sensors-22-00526-f008:**
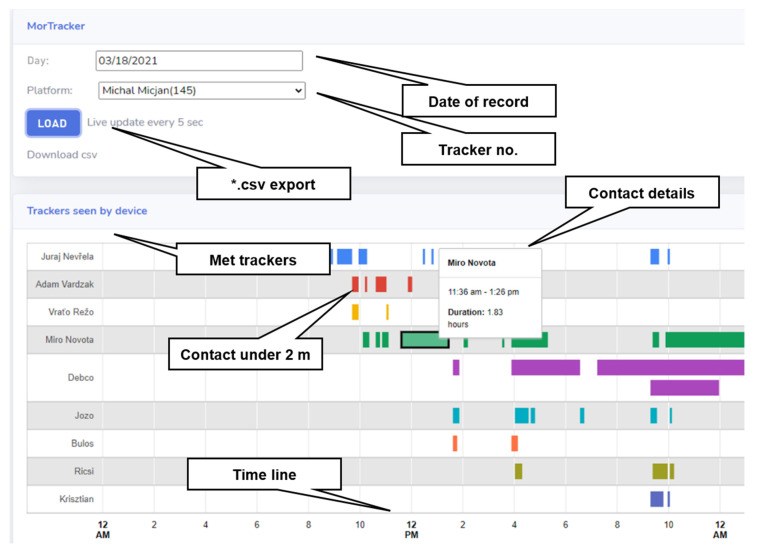
Functional testing—temporary user interface: Relevant (under 2 m) contacts of tracker no. 145 (Michal Micjan) on 18 March 2021 are displayed. Highlighted 1.83-h meeting with Miro Novota at 11:36 a.m.

**Figure 9 sensors-22-00526-f009:**
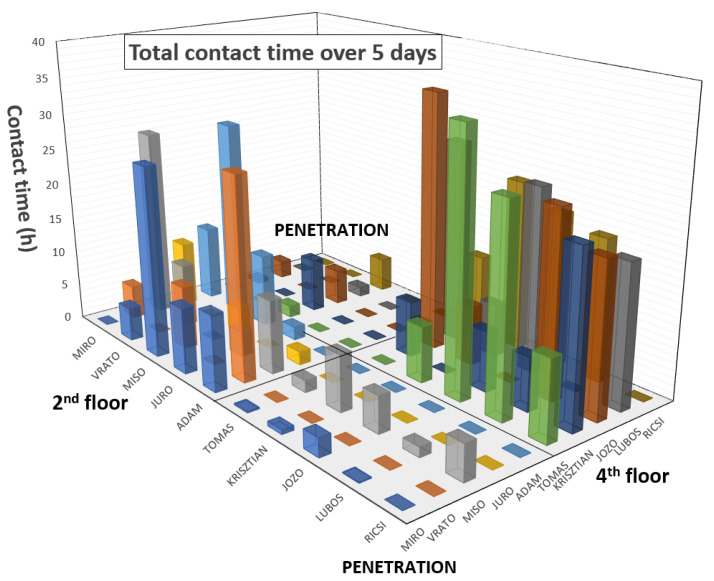
Functional testing—5-day contact time analysis: 2 separate working groups (2nd floor and 4th floor) and their cooperation (penetration).

**Figure 10 sensors-22-00526-f010:**
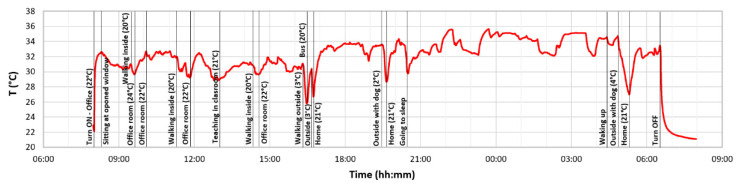
Functional testing—24-h peripheral temperature analysis with marked events and environmental temperatures.

**Table 1 sensors-22-00526-t001:** Tracker device parameters.

**Tracking sensor** Communication protocol RSSI Receiver sensitivity	**Laird BL654** Proprietary 2.4 GHz 1 dB resolution −95 dBm (1 Mbps)
**Microcontroller**	**Nordic nRF52840** 64 MHz Cortex-M4 with FPU 1 MB Flash, 256 KB RAM 2.4 GHz Transceiver
**Real-time clock** Accuracy	**Micro Crystal RV-8263-C7 RTC** ±20 ppm @ 25 °C
**Temperature sensor** Range/Accuracy Resolution Body contact	**Silicon Labs SI7051-A20-IM** 35.8–41 °C/±0.1 °C 20–70 °C/±0.13 °C 14-bit Surgical steel
**Barometer** Range Precision/Absolute accuracy	**Infineon Technologies DPS310** 300–1200 hPa ±0.002 hPa/±1 hPa
**Internal memory** Format Size Clock	**Macronix MX25R1635FZUILO** Flash 16 Mb 33 MHz
**Dimensions** (case)	Ø 46 × 13 mm
**Weight**	30 g (strap included)
**Power supply** Battery life Charging	3.7 V Li-Pol 300 mAh Approx. 80 days 60 min via a micro-USB connector
**Next features**	RGB LED signalization Switch on/off by unplugging/plugging to the charger Service button

**Table 2 sensors-22-00526-t002:** Detection gateway parameters.

**Microcontroller**	**Nordic nRF52840** 64 MHz Cortex-M4 with FPU 1 MB Flash, 256 KB RAM 2.4 GHz transceiver
**Tracker communication** Communication protocol Receive sensitivity Signal zone	**Laird BL654-PA extended with LNA** Proprietary 2.4 GHz −98.5 to −107 dBm Ideal: 1.6 km (outside)/Real: 50 m (inside)
**LTE** Data transmission Data rates	**Quectel EG912Y** LTE, GSM/GPRS/EDGE 10/5 Mbps (downlink/uplink @ LTE FDD)
**Wi-Fi** Wi-Fi protocol CPU	**Espressif ESP32 WROOM** 802.11 b/g/n (802.11n up to 150 Mbps), 2.4 GHz ESP32-D0WD-V3 Dual Core 240 MHz
**Antenna** Number × connector Peak gain	**Laird 2.4 GHz Dipole RF** 2 × SMA +2 dBi
**Internal memory** Format Size Clock	**Macronix MX25R6435FZNIL0** Flash 64 Mb 80 MHz
**Dimensions** (without antenna)	105 × 105 × 27 mm
**Weight** (without antenna)	100 g
**Power supply**	5 V/4 A via power jack

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
