# Peer review of "Research and Development of a COVID-19 Tracking System in Order to Implement Analytical Tools to Reduce the Infection Risk"

_sensors, 2022, doi:10.3390/s22020526_

Round 1
Reviewer 1 Report
The authors raise a very important issue related to counteracting and preventing the spread of covid-19 infections with the use of new technologies. In their work, the authors interestingly present the available methods of screening for potential sources of infection.
The proposed solution itself is very interesting and perhaps it would be very useful. However, I see a fundamental problem that is beyond the control of the authors. It is the reluctance of citizens to take any additional control of them. The proposed system only makes sense with indiscriminate use and this could only be implemented through appropriate legislation and enforcement in the form of penalties. Moreover, in view of the lack of universality, it carries with it the dangers of false security. The lack of an alarm means that the person is healthy, and in the absence of the tracker - false healthy. This can make infection easier by giving people in the system a false sense of security.
How did the authors solve the problem of interference on this 2.4Ghz frequency?
There are a lot of different applications that do similar things, also based on citizens' phones, but due to the distrust of their own authorities, they did not gain popularity.
However, this does not undermine the authors' achievements and their research and development work. However, it should be treated as an interesting idea with no chance of implementation.
Publishing wat article.
Author Response
Dear reviewer,
Thank you for your time and opinion. We incorporated your comments as best we could. We divided the report into points and processed each one. Changes in the text and our answers are written below:
The authors raise a very important issue related to counteracting and preventing the spread of covid-19 infections with the use of new technologies. In their work, the authors interestingly present the available methods of screening for potential sources of infection. The proposed solution itself is very interesting and perhaps it would be very useful.
Point 1: However, I see a fundamental problem that is beyond the control of the authors. It is the reluctance of citizens to take any additional control of them. The proposed system only makes sense with indiscriminate use and this could only be implemented through appropriate legislation and enforcement in the form of penalties.
Response 1: In the initial stages, of course, we do not calculate with the mass deployment of the tracking bracelet. As in several other countries, the facility is planned to be deployed primarily in places with a high concentrations of people and where it is impossible to trace contacts, such as large factories and mass social events. From our point of view, the current legislation in Slovakia, which will probably be also in other countries, is set even stricter. Without testing today, noone even get to work, shops, cannot participate in social events, etc. A home-office is ordered. Our equipment would rather mitigate the regulations, and thus provides another alternative solution. In some exposed areas, we must register to check the contacts so that the contacts can be traced. And our device actually does the same thing automatically with increased privacy protection. We are aware that priority regulations are handled by the government, but we are offering them an alternative tool. In companies, the bracelet can also be used in the form of a work safety regulation. We added text in the lines 72-76.
Point 2: Moreover, in view of the lack of universality, it carries with it the dangers of false security. The lack of an alarm means that the person is healthy, and in the absence of the tracker - false healthy. This can make infection easier by giving people in the system a false sense of security.
Response 2: Of course, we cannot guarantee wearing the bracelet completely. As we mentioned, we want to start using our equipment primarily in places where a lot of people gather and contacts are hard to find. It must be clearly stated to the wearer that the bracelet does not check its current state of infection in any way. It only tracks contacts and helps to find risky contacts. In the bracelet, the alarm (threat level) is displayed optically via RGB LED diodes. The user can also check the status using the web interface and if wearer pairs the bracelet with the phone, may also receive an SMS notification. The benefit for the wearer is that quickly notice that has been in contact with the infected person and can adapt accordingly. The device could contribute to the non-abolition of social life even under current conditions and regulations. Apparently we emphasized this weakly in the manuscript. Therefore, we have modified the text in the introduction (The lines 64-69). We have also modified Fig. 2 and 4. In Fig. 2 the device is photographed in the state with the alarm triggered „high“ and in Fig. 4 we added an output block with an alarm.
Point 3: How did the authors solve the problem of interference on this 2.4Ghz frequency?
Response 3: In the 2.4 GHz frequency range, Wi-Fi networks are the main source of interference. The reason we chose to use the 2.4 GHz proprietary protocol and not wifi or bluetooth is that uses different encoding and we can also change the wave modulation what minimize interference. The interference problem did not occur during the entire testing period in an environment with strong coverage of 2.4 GHz RF networks. Adde text in the lines 275-278, 535-537, 572-576.
Point 4: There are a lot of different applications that do similar things, also based on citizens' phones, but due to the distrust of their own authorities, they did not gain popularity. However, this does not undermine the authors' achievements and their research and development work. However, it should be treated as an interesting idea with no chance of implementation. Publishing wat article.
Response 4: There are a really large number of solutions to the problem of contact tracking. But our device has a few original solutions and benefits. The user is able to remain significantly more anonymous than when using a mobile phone, which is paired with a specific person. We also do not determine the user's location. We primarily look for a bracelet that shows the degree of threat. The doctor who inserts the alarm into the system does not know the other contacts and the health condition is known only to the specific patient and his doctor. Compared to mobile phones, we also have significantly longer battery life. We can operate for about 80 days on a single charge. Many people, especially the elderly, do not know how to use or do not have mobile phones. Our equipment is significantly cheaper. The employer also can't force people to install the software on their private phones. In many operations, the use of mobile phones or cameras is prohibited. A large part of the imaginary competing facilities lacks a sufficient description of the technical solution. We did not find during the search a single device that would use our chosen 2.4 GHz proprietary protocol. Due to it our device can be cheaper, more resistant to interference, we have a greater signal range, so we can solve the monitoring of mutual distance and communication with only one signal. We made a detection gateway, something like a wifi router, but just communicating on us set proprietary protocol. In parallel with the production of this tracker, we are also developing wearable health electronics, which has huge benefits at this time of COVID pandemic. We plan to combine these solutions into one whole and thus gain in further attractiveness for future users. We therefore believe that we can convince you and subsequent readers of the originality of our contribution. Added text in the lines 76-81, 560–576.
Thank you for your review and all the comments and remarks that we do believe helped improve the quality of the revised paper.
Reviewer 2 Report
This paper summarizes the development of a COVID-19 tracking system using proximity sensors allied with an if-then-else logic and a back-end database. The premise of the paper proposes to measure the temperature of the host using a wearable technology and comparing it with other wearable devices that are wi-fi enabled to check for heightened temperature and their proximity to known COVID-19 transmission. The proposed outcome of this system is a light-based notification similar to a traffic light. This paper only reports the proposal of the idea, the development of the sensor and limited UI details. There is a very short section on some results and some discussions on what it means. This is the primary weakness of this paper. Other than a section on Functional Testing, there is no other testing of the tracker that is reported. This calls into question a whole multitude of system, usability, and validity testing questions about the tracker itself. Moreover, similar trackers are further in the development, testing and reporting stages at the time of this review across multiple countries. My suggestion to improve this paper would be to report any testing that was done on this tracker in much more detail and to frame the manuscript as the findings of a study done in the two reported institutions in Slovakia. The findings can be a valuable tool in sharing knowledge against this virus globally.
Author Response
Dear reviewer,
Thank you for your time and opinion. We incorporated your comments as best we could. We divided the report into points and processed each one. Changes in the text and our answers are written below:
Moderate English changes required
We revised the English in the manuscript.
Point 1: This paper summarizes the development of a COVID-19 tracking system using proximity sensors allied with an if-then-else logic and a back-end database. The premise of the paper proposes to measure the temperature of the host using a wearable technology and comparing it with other wearable devices that are wi-fi enabled to check for heightened temperature and their proximity to known COVID-19 transmission. The proposed outcome of this system is a light-based notification similar to a traffic light.
Response 1: The priority task of our device is to track the mutual interaction of people, thus to measure the distance and length of personal contacts. The main effort is to record especially contacts between foreign people, so that in case of infection of one of them, also close unknown contacts can be found and warned. Temperature monitoring is a bonus and the temperature alarm is only local. We use 2.4 GHz proprietary protocol, which makes our device original. Wifi is used only in our other device, the detection gateway, and is only used to communicate with the backend, not the tracking devices themselves. Apparently, we presented the advantages of our product and the principle of operation a little weaker. Therefore, we proceeded to edit the texts in the introduction, discussion (Response 2) and priciple (The lines 266-269). Figures 2 and 4 have also been edited for this purpose.
Point 2: This paper only reports the proposal of the idea, the development of the sensor and limited UI details. There is a very short section on some results and some discussions on what it means. This is the primary weakness of this paper. Other than a section on Functional Testing, there is no other testing of the tracker that is reported. This calls into question a whole multitude of system, usability, and validity testing questions about the tracker itself. Moreover, similar trackers are further in the development, testing and reporting stages at the time of this review across multiple countries. My suggestion to improve this paper would be to report any testing that was done on this tracker in much more detail and to frame the manuscript as the findings of a study done in the two reported institutions in Slovakia. The findings can be a valuable tool in sharing knowledge against this virus globally.
Response 2: There are a really large number of solutions to the problem of contact tracking. But our device has a few original solutions and benefits. The user is able to remain significantly more anonymous than when using a mobile phone, which is paired with a specific person. We also do not determine the user's location. We primarily only look for a bracelet that shows the degree of threat. Compared to mobile phones, we also have significantly longer battery life. A large part of the imaginary competing facilities lacks a sufficient description of the technical solution. We did not find a single device that would use our chosen 2.4 GHz proprietary protocol. This protocol is cheaper, more resistant to interference, have a greater signal range, and we can solve the monitoring of mutual distance and communication with only one signal. We also made a detection gateway, something like a wifi router, but just communicating on this proprietary protocol. We consider our hardware (both tracker and detection gateway) to be the main benefit of the article. Therefore, we wanted to go out at this stage and not wait for the final completion of the software and detailed analysis of the results. In these difficult times, we want to save time and resent other scientists that tracking can also be done with this methodology. Added text in the lines 76-81, 275-278, 535-537 and 560-576.
We acknowledge that the mentioned UI and results are limited. Therefore, we added two images (Figure 7a and 7b) from our UI and a corresponding comment (The lines 441-447) in Chapter 4.3. And we also subjected contact tracking and daily temperature recording to a more detailed analysis. In Chapter 5. „Results“, we added two graphs (Figure 9 and 10) with the appropriate description and analysis of the results (The lines 466-509).
Thank you for your review and all the comments and remarks that we do believe helped improve the quality of the revised paper. We hope we have been able to meet your requirements
Reviewer 3 Report
This paper introduces a system for COVID tracking based on some existing tracking devices and detection gateways. The topic is important and authors are looking to extend the work in the future to include other human physiology like heart pulse. However, authors need to address the following issues before the paper is accepted:
- The submitted paper is clearly just a draft since it contains the language spelling check which had to be removed before submitting.
- Intensive language check is required.
- The abstract needs to be expanded to explain the methodology briefly.
- The 2.4 GHz proprietary protocol was not highlighted enough in the paper since it’s the essence of the work.
- Figure captions need to be expanded to fully explain each object in the figure.
- Shortcuts must be mentioned before the acronym like in LNA (Low-noise amplifier).
- It was mentioned that the number of packet loses is [64,65]. Can you explain more?
- How did you calculate the accuracy? It’s better to add some equations.
- Remove screenshots as they are not considered figures.
- How unique is your functional tracking system? Have you created the GUIs? Explain the contributions as you are using many existing hardware.
Author Response
Dear reviewer,
Thank you for your time and opinion. We incorporated your comments as best we could. We divided the report into points and processed each one. Changes in the text and our answers are written below:
Point 1: This paper introduces a system for COVID tracking based on some existing tracking devices and detection gateways. The topic is important and authors are looking to extend the work in the future to include other human physiology like heart pulse. However, authors need to address the following issues before the paper is accepted:
Response 1: In the article, we focused on the design of the original equipment, which we also built. The specific technical parameters are original and do not resemble any other device.
Point 2: The submitted paper is clearly just a draft since it contains the language spelling check which had to be removed before submitting.
Response 2: The version you received is not the original. It is version 2 after the first review. The "tracking changes” and "spelling check" in the document is a MDPI prescribed procedure. Changes from version 2 to version 3 will now be highlighted as well. Of course, we will be happy to send you a version without tracking enabled.
Point 3: Intensive language check is required.
Response 3: We have revised formal errors and language in our article.
Point 4: The abstract needs to be expanded to explain the methodology briefly.
Response 4: We expanded the methodology part in the abstract.
Point 5: The 2.4 GHz proprietary protocol was not highlighted enough in the paper since it’s the essence of the work.
Response 5: We added sentences in lines 89-102. Further text on this issue can be also found on the line 305-315, 401-408, 622-634 and 662-668.
Point 6: Figure captions need to be expanded to fully explain each object in the figure.
Response 6: We added simple object explanations in all the figure captions. More details can be found in text.
Point 7: Shortcuts must be mentioned before the acronym like in LNA (Low-noise amplifier).
Response 7: We revised and controlled all abbreviations in the article. Acronym LNA was mentioned on line 406.
Point 8: It was mentioned that the number of packets loses is [64,65]. Can you explain more?
Response 8: Added text on line 414-418. “Packet loss describes packets of data not reaching their destination after being transmitted across a network. In our system, we know the number of packets sent per second. Packet loss is in our case majority depending on the distance and location of the people. By combining two parameters, we increase the reliability of the system.”
Point 9: How did you calculate the accuracy? It’s better to add some equations.
Response 9: If you think the accuracy of distance measurement by trackers. This was done by empirical calibration (Line 609-612). If you mean the accuracy of the circuits (temperature, barometer, RTC, etc.), they were based on catalog datasheets. As for 97% match of tracker records, this was done by comparing the time-lapse data of the two relevant trackers.
Point 10: Remove screenshots as they are not considered figures.
Response 10: I assume you mean Fig. 7 and we agree with you. In the original version they were not, they are added based on the request from the first review round. The editor will probably have to decide on this. I have now upgraded them, so that they are not just plain print screens, but also provide additional information.
Point 11: How unique is your functional tracking system? Have you created the GUIs? Explain the contributions as you are using many existing hardware.
Response 11: The most unique is the use of a 2.4 GHz proprietary protocol. Detailed benefits can be found in the Discussion in lines 622 – 686. The only existing hardware parts were purchased electronic components. Everything else is our own original work. Hardware (tracking devices and gateways), front-end and GUI too. The front-end is a combination of Php, AngularJS and Phtml. The back-end is made in Python with a kernel in C ++ and a database in PostgreSQL. The tracking device has been granted a patent.
Round 2
Reviewer 2 Report
Sections 4 and 5 are still lacking. There are insufficient details pertaining to the testing that was carried out. There are almost no details about what was carried out and what was found. There are no measurable outcomes or use case details that would be warranted for any type of functional testing. Indicated in the paper is data collected from 15 colleagues. This is considered human testing. Lacking is any ethics board permissions to allow such testing to be carried out. There are no experiment design details to allow the reader to understand what is being demonstrated in the UI that is presented.
Author Response
Dear reviewer,
Thank you for your time and opinion.
Point 1: Sections 4 and 5 are still lacking. There are insufficient details pertaining to the testing that was carried out. There are almost no details about what was carried out and what was found. There are no measurable outcomes or use case details that would be warranted for any type of functional testing.
Response 1: Our main result was the design of the original hardware and its quick introduction to other scientists, as a possible hardware solution that has several advantages. Further results will be available only after the equipment is put into real operation. We did the best we could in this phase. We performed a 24-hour temperature recording with a bracelet, calculated the total interaction of 10 people in two working groups, and presented the developed UI. Unfortunately, we are no better able to meet your additional requirements. The concept of our article is built differently.
Point 2: Indicated in the paper is data collected from 15 colleagues. This is considered human testing. Lacking is any ethics board permissions to allow such testing to be carried out. There are no experiment design details to allow the reader to understand what is being demonstrated in the UI that is presented.
Response 2: We have added the wording in the text. All participants were informed and they signed an informed consent.
Reviewer 3 Report
Thank you for addressing the issues.